# Importance of Human Breast Milk in the Early Colonization of *Streptococcus mutans*

**DOI:** 10.3390/medicina60081308

**Published:** 2024-08-13

**Authors:** Karina Córdova-Carrillo, Cristina De la Peña-Lobato, María Verónica Cuevas-González, Juan Carlos Cuevas-González, León Francisco Espinosa-Cristóbal, Karla Lizette Tovar-Carrillo, Rosa Alicia Saucedo-Acuña, Graciela Zambrano-Galván, Simón Yobanny Reyes-López

**Affiliations:** 1Institute of Biomedical Sciences, Autonomous University of Ciudad Juarez, Juarez City 32310, Mexico; al220678@alumnos.uacj.mx (K.C.-C.); cristina.delapena@uacj.mx (C.D.l.P.-L.); maria.cuevas@uacj.mx (M.V.C.-G.); leon.espinosa@uacj.mx (L.F.E.-C.); karla.tovar@uacj.mx (K.L.T.-C.); rosauced@uacj.mx (R.A.S.-A.); simon.reyes@uacj.mx (S.Y.R.-L.); 2Research Division, Faculty of Medicine, Juarez University of the State of Durango, Durango City 34113, Mexico; gzambrano@ujed.mx

**Keywords:** early colonization, *Streptococcus mutans*, human breast milk

## Abstract

*Background and objectives*: The development of the oral microbiome begins in the prenatal stage. Breast milk contains antimicrobial proteins, microorganisms, metabolites, enzymes, and immunoglobulins, among others; therefore, differences have been noted in the type of microorganisms that colonize the oral cavity of children who are breastfed compared to those who are formula-fed. Our objective was to establish the relationship between breastfeeding, formula feeding, or mixed feeding (breastfeeding and formula) with the presence of *S. mutans* in a population of children under 6 months of age. *Materials and Methods*: The patients were recruited from the Child Care Center of Ciudad Juárez, Chihuahua, and from the pediatric dentistry postgraduate clinics of the Autonomous University of Ciudad Juárez; children exclusively fed maternally, with formula, and/or mixed were included. Those who had been fed within the previous hour were excluded. The sample was taken with a smear of the jugal groove using a sterile micro-brush. For the identification of *Streptococcus mutans*, a culture of Mitis Salivarius Agar (Millipore) was used. *Results*: 53.3% corresponded to females and 46.7% to males, 36.7% corresponded to maternal feeding, 23.3% corresponded to formula feeding, and 40% corresponded to mixed feeding. In 90% of the infants, the parents indicated that they did not perform oral hygiene. The CFU count showed that infants who were exclusively breastfed had an average of 9 × 10 CF/mL, formula-fed infants had an average of 78 × 10 CFU/mL, and those who had mixed feeding 21 × 10 CFU/mL. *Conclusions*: According to the results obtained, it was possible to corroborate that exclusive breastfeeding limits the colonization of *Streptococcus mutans* compared to those infants who receive formula or mixed feeding; these results could have a clinical impact on the dental health of infants by having a lower presence of one of the main etiological factors involved in dental caries and the type of microbiome established in the oral cavity.

## 1. Introduction

*Streptococcus mutans* (*S. mutans*) is one of the main pathogens related to the development of caries. This pathogen is characterized by its high adherence to the surface of teeth, acid production, and its ability to survive in acidic environments [1]. It is an obligate human pathogen; that is, it is not free-living. It is a non-sporulating Gram-positive bacteria. *S. mutans* is composed of approximately 2000 genes, about 500 of which can present with mutations, giving it pathogenic characteristics that may influence the susceptibility of individuals in relation to the development of caries [2].

Since the discovery of the microbiome, it has been the subject of research analyzing its diversity and the impact it has on general health. In various research, it has been suggested that the establishment of the microbiome begins in intrauterine life. After birth, the microbiome changes due to the contact that the infant has with the environment, their mother, and their family through skin-to-skin contact and kisses, but mainly through breastfeeding. This is key to establishing the intestinal and oral microbiome [3,4].

The development of the oral microbiome specifically begins in the prenatal stage; microorganisms, such as Firmicutes, Tenericutes, Proteobacteria, Bacteroidetes, and Fusobacteria phyla, have been identified in the placenta and are related to the mother’s oral microbiome [1]. The moment of delivery is another relevant stage since the route through which it is carried out determines the initial diversity of microorganisms; babies born through the vaginal canal present greater bacterial heterogeneity in the oral cavity compared to those born via cesarean section [2]. Within the initial days after birth, the microbiome is highly dynamic due to interactions with the mother, the family, the hospital environment, and feeding, allowing for an adequate stimulation of both innate and acquired immunity, influencing health–disease states in future life stages [5].

The feeding that newborns receive plays a key role in the establishment of the microbiome. Breast milk possesses antimicrobial proteins, microorganisms, metabolites, enzymes, immunoglobulins, etc. Therefore, differences have been demonstrated in the type of microorganisms that colonize the oral cavity of children who are breastfed compared to those who are formula-fed [6]. Among the most abundant microorganisms in the oral cavity and the main one responsible for carious lesions is *S. mutans*, which is acquired from the first days of life, mainly via vertical transmission through the mother and/or family members [7]. The early presence of this microorganism entails a risk factor for the development of caries lesions in primary dentition, especially when combined with the absence of oral hygiene. In our population, there have been no studies to analyze the impact that breastfeeding has on the establishment of streptococcus mutans in infants under 6 months of age; therefore, the objective of this study is to establish the relationship between breastfeeding, formula feeding, or mixed feeding (breast and formula feeding) with the presence of *S. mutans* in a population of infants under 6 months of age attending day care in Ciudad Juárez, Chihuahua. A non-probabilistic comparative observational descriptive study was carried out.

## 2. Materials and Methods

This is a non-probabilistic comparative observational descriptive study in which patients were recruited from the Child Care Center of Ciudad Juárez, Chihuahua, and from the pediatric dentistry postgraduate clinics of the Universidad Autónoma de Ciudad Juárez (approved by the institution’s bioethics committee number CEI-2023-2-997), with prior informed consent being obtained from the parents and/or legal guardians. Convenience sampling was conducted. The selection criteria were children with exclusive breastfeeding, formula, and/or mixed feeding under 6 months of age. The exclusion criteria included children with complementary feeding of any form, children who had been fed within the previous hour, children who had taken antibiotics in the month prior to sample collection, and finally, children who were already in the process of dental eruption and those who presented any clinical symptoms of disease.

For sample collection, a clinical examination of the oral cavity was carried out to ensure that there were no lesions, afterwards, the oral cavity was cleaned with a wet gauze pad, and after 15 min, a smear of the attached posterior gingiva was made with a sterile micro-brush. The sample was subsequently placed in 1 mL of sterilized distilled water. The sample was then placed on ice and transported to the laboratory. All of the samples were processed on the same day that they were taken.

For the identification of *S. mutans*, a cell culture of Mitis Salivarius Agar (Millipore) was used, and the Petri dishes were prepared according to the manufacturer’s instructions. For this, 100 µL of the sample was swabbed and incubated for 24 h at 5% CO_2_ and 37 °C. For the identification of *S. mutans*, elevated, convex, wavy, opaque, pale blue, granular-appearing colonies were taken as parameters. The counting of colony-forming units (CFUs) was carried out using a digital colony display (VECVP-CM3) by two evaluators; the device had previously been calibrated with a Kappa of 0.90. In case of discrepancy, a third experiment evaluator participated. All of the samples were assessed in triplicate.

A statistical analysis was carried out using the SPSS V.22 program, and a descriptive analysis of the variables was performed. To establish an association between the qualitative variables, X2 or Fisher’s exact test was carried out, depending on the distribution of the sample. To analyze the association between the study groups, a Mann–Whitney or Kruskal–Wallis U test was performed, depending on the distribution of the data.

## 3. Results

A total of 30 samples were collected, of which 53.3% corresponded to females and 46.7% to males; 36.7% corresponded to breastfeeding, 23.3% to formula feeding, and 40% to mixed feeding. In 90% of the infants, the parents indicated that they did not perform oral hygiene on the infant (Table 1).

To count the CFUs, the samples were divided into three groups according to the type of feeding: exclusive maternal feeding, formula feeding, and mixed feeding (maternal and formula). The CFU count showed that infants who were exclusively breastfed had an average of 8.5 × 10 CFU/mL (σ = 15.03 × 10 CFU/mL), formula-fed infants had an average of 77.7 × 10 CFU/mL (σ = 100.63 × 10 CFU/mL), and mixed feeding infants had an average of 20.5 × 10 CFU/mL (σ = 28.7 × 10 CFU/mL) (Figure 1). When analyzing the association between the variables, statistical significance was identified between oral hygiene and feeding type (*p* = 0.04), as shown in Figure 1.

Finally, when analyzing the comparison of the CFU average with the type of feeding, a significant difference was observed (*p* = 0.04), demonstrating the difference between the type of feeding and the presence of *S. mutans* (Table 2).

## 4. Discussion

Childhood caries are considered a worldwide public health problem. The latest studies indicate that approximately 48% of children experience caries [8], and the presence of *S. mutans* is one of the main microorganisms related to the disease [9] because of its ability to produce acids as a result of carbohydrate fermentation. As mentioned before, *S. mutans* is present in the oral cavity since the first days after birth, with the main entry route being the contact that infants have with their family through kisses or through a pacifier, which provides an ideal environment for the growth of *S. mutans*, resulting in its permanence in the oral cavity, even without the presence of teeth [10].

Multiple studies have demonstrated the impact of breastfeeding on the development of oral microflora; however, the relationship that this type of feeding directly and specifically has with *S. mutans* is not yet well determined. Oba PM et al. analyzed the microbiome of infants in relation to solid food, formula feeding, and breastfeeding and demonstrated differences in the type of microorganisms present, reporting that breastfed infants have high concentrations of Streptococcus [11]. On the other hand, García-Quintana A et al. carried out a bacterial culture study focusing on two-month-old infants, comparing the type of bacteria and the type of feeding, demonstrating that Firmicutes phylum (Streptococcus, Lactobacillu and Staphylococcus) predominates in terms of maternal nutrition [12]. Our results highlight that children who receive maternal feeding have a lower presence of *S. mutans* compared to other types of feeding. The differences between our results and previous studies are mainly due to two factors: the method used for bacterial identification and the size of the sample.

The components of breast milk, like proteins, lipids, carbohydrates, and microorganisms, are key elements for human growth and development. The microorganisms present in human breast milk have the function of providing neonates with their own microbiota, as well as stimulating their defense system. Breast milk contains around 800 different species of bacteria, the most common being facultative anaerobes or strict aerobes. Although the type of microorganism varies depending on the mother’s age, diet, and exposure to antibiotics, *Staphylococcus* and *Streptococcus* are considered elemental microorganisms in breast milk [13,14].

According to the results obtained in this study, the presence of *S. mutans* was significantly lower in children with exclusive maternal feeding compared to formula or mixed feeding, and the possible explanation for this event is that maternal milk contains anticariogenic elements, such as casein, whey protein, lactose, and milk fat. Casein inhibits the adherence of *S. mutans*. Another component of breast milk, lactoferrin, has antimicrobial and antioxidant effects [15,16]. Although breastfeeding is considered a protective factor against early caries, it has been documented that when breastfeeding lasts more than 12 months with a daily frequency of seven times, it increases the risk of infant caries due to the high glucose content of breast milk (6.4–7.6 g/dL) [16] Van Meijeren-van et al. carried out a very interesting cohort study to evaluate the relationship between breastfeeding and the risk of caries development in relation to the socioeconomic level, and the result showed that children who were breastfed for more than 12 months had a significant increase in caries development, with an OR of 1.35 and 95% CI 1.04–1.74. On the other hand, no association was found between caries development and maternal nutrition during the first 4 months of life in children above the age of 6 years old [17].

Additionally, the immunological component of breast milk is also relevant, integrating the importance of maternal nutrition into the prevention of childhood caries. The colostrum produced from birth until approximately day 7 contains high amounts of immunoglobulins, leukocytes, vitamins, hormones, etc., and around 80% of the leukocytes correspond to macrophages that migrate from the blood mammary epithelium. These cells differentiate into Dendritic cells, stimulating the activity of T lymphocytes and protecting against pathogens. In addition to this, cytokines such as TNFα, IL-6, IL-8, and IFNγ have also been identified, which contribute to regulating bacterial colonization [18,19]. On the other hand, the immunoglobulins present in breast milk help maintain bacterial balance and stimulate the defense system. The presence of sIgA inhibits the growth and adherence to epithelial cells of *S. mutans* as well as its metabolic activity [20,21].

Finally, other factors that contribute to the establishment of the microbiome are hygiene habits. From our experience, mothers or caregivers do not usually perform oral hygiene until the eruption of dental organs begins, and this was demonstrated in the results shown in our study, where 90% of children’s parents do not perform any type of oral hygiene. Finally, the level of education of the mother is of great relevance since it has been shown that higher levels of education correlates with a greater probability of breastfeeding the baby within the first hours; this was demonstrated by Acharya P et in his work, in which the educational level of the mother in relation to early maternal feeding was evaluated, reporting an OR of 1.63 [22]. In our work, this information was not possible to corroborate because the childcare center from which the samples were taken is focused on providing care to working mothers of the Ministry of Public Education; therefore, all mothers share the same educational level.

Although this study corroborates the importance of maternal nutrition in the development of a cariogenic oral microbiome, there were multiple limitations, such as the very limited study population, the non-specific identification method used in relation to *S. mutans*, as well as the lack of control over the food that the infants received from their caretakers; however, it lays the foundation for subsequent studies to be carried out with a much stricter methodology that will allow us to have solid data to create strategies to promote breastfeeding orientation towards oral health in our population. Likewise, it will allow us to propose studies with longer follow-up times, enabling us to corroborate the effect of breast milk on the development of carious lesions in children with complete primary dentition.

## 5. Conclusions

The early establishment of Streptococcus mutans is undoubtedly a risk for the development of early caries. The type of diet consumed within the first 6 months is of vital importance because it has been shown to impact the type of microbiome established in the oral cavity. According to the results obtained, it was possible to corroborate that exclusive breastfeeding limits the colonization of *S. mutans* compared to those infants who receive formula or mixed feeding. These results could have a clinical impact on the dental health of infants by having a lower presence of one of the main etiological factors of dental caries; therefore, the great importance of maternal nutrition is corroborated not only by the impact it has on the immune system but also by the impact on the establishment of the oral flora in regard to protection against early caries in conjunction with oral hygiene.

## Figures and Tables

**Figure 1 medicina-60-01308-f001:**
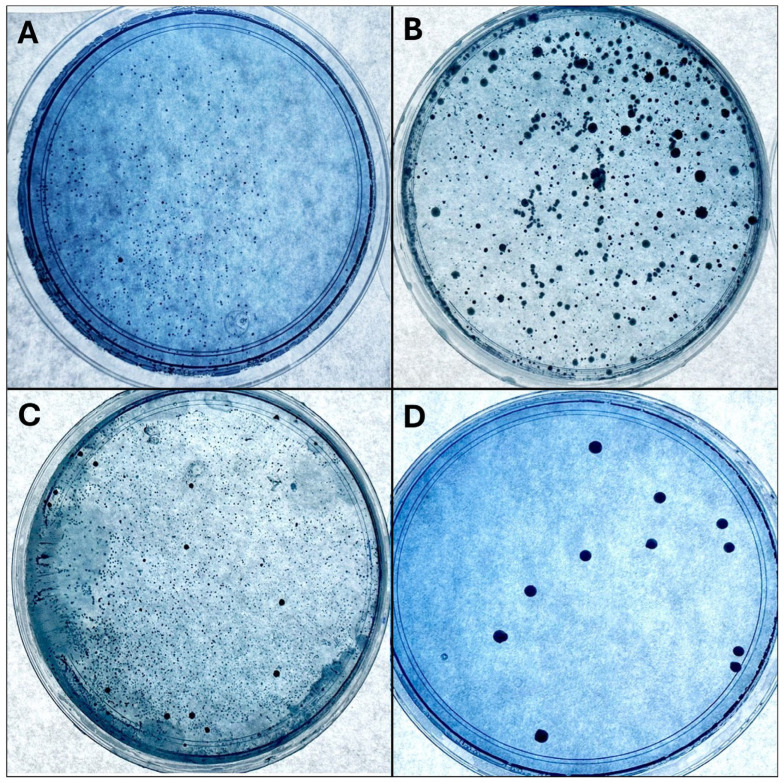
Bacterial cultures of the different samples are shown: (**A**) breastfeeding sample, (**B**) formula feeding sample, (**C**) mixed feeding sample, and (**D**) *S. mutans* positive control.

**Table 1 medicina-60-01308-t001:** Descriptive statistics of the cases included in the study.

Variable	*n* (%)
Sex	
Female	16 (53.3)
Male	14 (46.7)
Type of feeding	
Breastfeeding	11 (36.7)
Formula feeding	7 (23.3)
Mixed feeding	12 (40.0)
Presence of infections	
No	27 (90.0)
Yes	3 (10.0)
Medications taken previous month	
No	17 (56.7)
Anti-inflammatory	1 (3.3)
Antibiotic	3 (10.0)
Other	9 (30.0)
Personal medical history	
No	28 (93.3)
Yes	2(6.7)
Birth weight	
Low weight	3 (10)
Proper weight	24 (80)
High weigth	3 (10)
Complications at birth	
No	24 (80)
Yes	6 (20)
Gestation weeks	
Preterm	3 (10)
Full Term	26 (86.7)
Postterm	1 (3.3)
Type of delivery	
Natural Birth	13 (43.3)
Cesarean	17 (56.7)
Mother’s academic degree	
Elementary School	1 (3.3)
High School	2 (6.7)
Bachelor’s Degree	25 (83.2)
Postgraduate	2 (6.7)
Oral hygiene	
No	27 (90)
Yes	3 (10)
Colony-forming units	
Breastfeeding	9 × 10/mL (5 × 10 CFU/mL–53 × 10 CFU/mL)
Formula feeding	78 × 10 CFU/mL (6 × 10 CFU/mL–286 × 10 CFU/mL)
Mixed feeding	21 × 10 CFU/mL (2 × 10 CFU/mL–106 × 10 CFU/mL)

**Table 2 medicina-60-01308-t002:** Analysis of variables association through the X^2^ test.

Variable	Breastfeeding	Formula Feeding	Mixed Feeding	*p* (*X*^2^)
Sex				
Male	6	3	5	0.804
Female	5	4	7	
Previous month’s medications				
Yes	6	2	5	0.549
No	5	5	7	
Personal medical history				
Yes	2	0	0	0.157
No	9	7	12	
Birth weight				
Low	0	1	2	0.133
Proper	8	6	10	
High	3	0	0	
Birth complications				
Yes	2	2	2	0.808
No	9	5	10	
Gestation week				
Preterm	1	1	1	0.472
Full Term	9	6	11	
Postterm	1	0	0	
Type of delivery				
Vaginal	4	4	5	0.679
Cesarean	7	3	7	
Mother’s academic degree				
Elementary school	1	0	0	0.074
High school	0	2	0	
Bachelor’s Degree	10	5	10	
Postgraduate	0	0	2	
Oral hygiene				
Yes	0	3	0	0.004
No	11	4	12	

## Data Availability

The original contributions presented in the study are included in the article, further inquiries can be directed to the corresponding author.

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
