# Peer review of "Importance of Human Breast Milk in the Early Colonization of Streptococcus mutans"

_medicina, 2024, doi:10.3390/medicina60081308_

Round 1

Reviewer 1 Report

Comments and Suggestions for Authors

Dear Authors,

 The topic of the presented article is very interesting, however, I have many comments.

·       The same affiliation does not have to be written so many times. Just enter it once and add the appropriate number in the upper index for people associated with the same institution.

·       Nomenclature related to microorganisms does not include italics.

·       Can microbial results be reported as CFU/mL?

·       "those who had no type of complementary feeding or who had less than an hour since their last feeding were excluded" - this sentence would imply that those children who were exclusively breastfed (because they did not have any complementary feeding) were not included.

·       "Keywords: ...streptococcus Mutans..." - wrong capital and lowercase letters, no italics.

Introduction:

·       There is no need to mention Antony Van Leeuwenhoek here.

·       It is worth mentioning the importance of skin-to-skin contact after childbirth.

·       Is S. mutans really transmitted from mother to child by vertical transmission?

·       In the introduction, it is worth presenting the discussed microorganism in more detail.

Material and methods:

·       Other microorganisms also grow on the indicated microbiological medium. Was belonging to S. mutans assessed only on the basis of the appearance of bacterial colonies?

 Results:

·       Graphic refinement of the tables will be useful. Some headings are not evenly spacer.

Discussion:

·       In the discussion, it is worth citing more research on the importance of breastfeeding in preventing caries, or on the contrary, intensifying this health problem.

·       „in our results it is highlighted that children with maternal feeding have a low presence of S. mutans in comparison with other types of feeding, these differences are mainly due to two factors, the method used for bacterial identification as well as the size of the sample.” - This text indicate that the type of feeding the child and the method of conducting the tests do not matter. This was probably not the authors' intention.

Comments on the Quality of English Language

I think that the article requires language corrections. Many linguistic irregularities can be found here. Some sentences are ungrammatically linked together. They can be divided into two separate ones.

Author Response

Dear Review

We appreciate the time taken and the interest in reviewing our work entitled “Importance of human breast milk in the early colonization of the Streptococcus mutans” as well as the suggestions made in order to improve our work, for which each of your opinions was followed up promptly.

  • The same affiliation does not have to be written so many times. Just enter it once and add the appropriate number in the upper index for people associated with the same institution.

Answer.- The affiliation was changed as suggested

  • Nomenclature related to microorganisms does not include italics.

Answer.- The nomenclature of microorganisms was corrected

  • Can microbial results be reported as CFU/mL?

Answer.- The results were unified to CFU/mL

  • "those who had no type of complementary feeding or who had less than an hour since their last feeding were excluded" - this sentence would imply that those children who were exclusively breastfed (because they did not have any complementary feeding) were not included.

Answer. - The sentence was rewritten

  • "Keywords: ...streptococcus Mutans..." - wrong capital and lowercase letters, no italics.

Answer.- Were made The suggested changes

Introduction:

  • There is no need to mention Antony Van Leeuwenhoek here.

Answer.- The name of Antony Van Leeuwenhoek was removed from the text

  • It is worth mentioning the importance of skin-to-skin contact after childbirth.

Answer.- Was added a paragraph emphasizing the importance of skin-to-skin contact in newborn's microbiome establishing

  • Is S. mutans really transmitted from mother to child by vertical transmission?

Answer.- Recently studies support the theory of vertical transmission of the microbiome from mother to child, mainly during childbirth and breastfeeding

  • In the introduction, it is worth presenting the discussed microorganism in more detail.

Answer. - A paragraph was added mentioning the characteristics of S. mutans

Material and methods:

  • Other microorganisms also grow on the indicated microbiological medium. Was belonging to S. mutans assessed only on the basis of the appearance of bacterial colonies?

Answer. - That is right, the culture medium used only allows identification through the morphology of the bacteria, this information is specified in the methodology and in the weaknesses of the study in the discussion.

 Results:

  • Graphic refinement of the tables will be useful. Some headings are not evenly spacer.

Answer. - The graph was reviewed, and errors were corrected.

Discussion:

  • In the discussion, it is worth citing more research on the importance of breastfeeding in preventing caries, or on the contrary, intensifying this health problem.

Answer. - References were added to the discussion that emphasize the importance of breastfeeding

  • in our results it is highlighted that children with maternal feeding have a low presence of S. mutans in comparison with other types of feeding, these differences are mainly due to two factors, the method used for bacterial identification as well as the size of the sample.” - This text indicate that the type of feeding the child and the method of conducting the tests do not matter. This was probably not the authors' intention.

Answer.- the text was re-written

I think that the article requires language corrections. Many linguistic irregularities can be found here. Some sentences are ungrammatically linked together. They can be divided into two separate ones.

Answer.- The text was sent for grammatical review by an expert in the area.

Reviewer 2 Report

Comments and Suggestions for Authors

The manuscript in question is interesting and aims to establish the relationship between breastfeeding, formula feeding or mixed feeding (breastfeeding and formula) and the presence of S. mutans in a population of children under 6 months of age.

Some points must be considered before final publication:

The conclusions in the abstract seem very general and speculative. Point out the main outcomes of your work.

The introduction is short, explores the topic in a very superficial way, cites only five previous studies and is not fluid. The authors are invited to rewrite the section, providing a brief contextualization of the theme, explaining the gaps present in the literature on this subject and making the objective and reason for the study clearer, considering that the presence of Streptococcus mutans and the cariogenic properties of this microorganism are already well established. What's new here?

In the materials and methods section, authors should further detail the inclusion and exclusion criteria, describe the sample collection procedure in more detail, as well as their transportation and processing. Provide additional details on the method of identifying and counting CFUs.

In the summary, the "7.45 x 10 ml" value for CFUs in breastfed infants appears inconsistent with the value in the "8.5 x 10 ml" results section. Authors must ensure that all numerical data is consistent throughout the manuscript.

In the discussion section, the authors do not adequately compare the results with those of previous studies. I suggest developing a more robust discussion that compares the results in more detail with previous studies.

Discuss in more detail the limitations of the study and how they may have influenced the results and their implications. Also provide directions for future research.

In the conclusion section, authors should only reinforce the main findings of the study and their implications. What is the main discovery and its clinical implication?

Author Response

Dear Reviewer

  • The conclusions in the abstract seem very general and speculative. Point out the main outcomes of your work.

Answer.- the conclusion was modified

  • The introduction is short, explores the topic in a very superficial way, cites only five previous studies and is not fluid. The authors are invited to rewrite the section, providing a brief contextualization of the theme, explaining the gaps present in the literature on this subject and making the objective and reason for the study clearer, considering that the presence of Streptococcus mutans and the cariogenic properties of this microorganism are already well established. What's new here?

Answer.- Text was added to reinforce the importance of the study

  • In the materials and methods section, authors should further detail the inclusion and exclusion criteria, describe the sample collection procedure in more detail, as well as their transportation and processing. Provide additional details on the method of identifying and counting CFUs.

Answer.- The text was corrected and specific information were added about: selection criteria, sample collection methods and their analysis method.

  • In the summary, the "7.45 x 10 ml" value for CFUs in breastfed infants appears inconsistent with the value in the "8.5 x 10 ml" results section. Authors must ensure that all numerical data is consistent throughout the manuscript

Answer.- CFU values ​​were reviewed throughout the text.

  • In the discussion section, the authors do not adequately compare the results with those of previous studies. I suggest developing a more robust discussion that compares the results in more detail with previous studies

Answer.- References were added to complement an adequate discussion of the results

  • Discuss in more detail the limitations of the study and how they may have influenced the results and their implications. Also provide directions for future research.

 Answer.- Information about possible directions of research line were added.

  • In the conclusion section, authors should only reinforce the main findings of the study and their implications. What is the main discovery and its clinical implication?

Answer.- The conclusion was written in order to make it clearer.

Round 2

Reviewer 1 Report

Comments and Suggestions for Authors

Dear Authors,

Thank you for taking into account the comments sent to the article. I have noticed some inaccuracies:

·        „In our population, there are no studies that analyze the impact that breastfeeding has on the establishment of streptococcus mutans in infants under 6 months of age…” - there are still errors in the nomenclature of bacteria - lower case, no italics.

·        The number of microorganisms should be expressed as a number with one decimal place, it must be rounded according to the rules of mathematics and expressed as an exponent.

·        The phrase „maternal nutrition” may erroneously indicate the mother's nutrition/women's diets, when it is about feeding children.

Comments on the Quality of English Language

 Minor editing required.